# DetecDiv, a generalist deep-learning platform for automated cell division tracking and survival analysis

Théo Aspert[1,2,3,4]*, Didier Hentsch[1,2,3,4], Gilles Charvin[1,2,3,4]*

[1]Department of Developmental Biology and Stem Cells, Institut de Génétique et de Biologie Moléculaire et Cellulaire, Strasbourg, France; [2]Centre National de la Recherche Scientifique, Strasbourg, France; [3]Institut National de la Santé et de la Recherche Médicale, Strasbourg, France; [4]Université de Strasbourg, Strasbourg, France

**Abstract** Automating the extraction of meaningful temporal information from sequences of microscopy images represents a major challenge to characterize dynamical biological processes. So far, strong limitations in the ability to quantitatively analyze single-cell trajectories have prevented large-scale investigations to assess the dynamics of entry into replicative senescence in yeast. Here, we have developed DetecDiv, a microfluidic-based image acquisition platform combined with deep learning-based software for high-throughput single-cell division tracking. We show that DetecDiv can automatically reconstruct cellular replicative lifespans with high accuracy and performs similarly with various imaging platforms and geometries of microfluidic traps. In addition, this methodology provides comprehensive temporal cellular metrics using time-series classification and image semantic segmentation. Last, we show that this method can be further applied to automatically quantify the dynamics of cellular adaptation and real-time cell survival upon exposure to environmental stress. Hence, this methodology provides an all-in-one toolbox for high-throughput phenotyping for cell cycle, stress response, and replicative lifespan assays.

**\*For correspondence:**
theo.aspert@gmail.com (TA);
charvin@unistra.fr (GC)

**Competing interest:** The authors declare that no competing interests exist.

## Editor's evaluation

In this work, the authors describe a novel method, based on deep learning, to analyze large numbers of yeast cells dividing in a controlled environment. The method builds on existing yeast cell trapping microfluidic devices that have been used for replicative lifespan assay. The authors demonstrate how an optimized microfluidic device can be coupled with deep learning methods to perform automatic cell division tracking and single cell trajectories quantification. The overall performance of the method is impressive: it allows to deal with large image datasets generated by timelapse microscopy several order of magnitudes faster than what manual annotation would require.

## Introduction

Epigenetic processes that span several division cycles are ubiquitous in biology and underlie essential biological functions, such as cellular memory phenomena (*Caudron and Barral, 2013*; *Bheda et al., 2020*; *Kundu et al., 2007*), differentiation, and aging (*Denoth Lippuner et al., 2014*; *Janssens and Veenhoff, 2016*). In budding yeast, mother cells undergo about 20–30 asymmetric divisions before entering senescence and dying (*Mortimer and Johnson, 1959*). Over the last decades, this simple unicellular has become a reference model for understanding the fundamental mechanisms that control longevity (*Denoth Lippuner et al., 2014*; *Janssens and Veenhoff, 2016*).

Several independent mechanistic models have been proposed to explain entry into replicative senescence, including asymmetric accumulation of extrachromosomal rDNA circles (ERCs) (*Sinclair and Guarente, 1997*), protein aggregates (*Aguilaniu et al., 2003*), signaling processes associated with loss of vacuole acidity (*Hughes and Gottschling, 2012*), or loss of chromatin silencing (*Pal and Tyler, 2016*). Classical replicative lifespan (RLS) assays by microdissection, combined with genetic perturbations, have been decisive in identifying and characterizing genetic factors and pathways that influence longevity in budding yeast (*McCormick et al., 2015*). Similarly, enrichment techniques of aged mother cells in a batch provided further understanding of the physiology of cellular senescence in this model organism (*Lindstrom and Gottschling, 2009*; *Janssens et al., 2015*).

However, how the appearance of markers of aging is coordinated temporally and causally remains poorly understood (*Dillin et al., 2014*; *He et al., 2018*). In part, this is due to the difficulty of directly characterizing the sequence of events that constitute the senescence entry scenario: RLS assays by microdissection generally give no information other than the replicative age upon cell death; old cells enrichment techniques ignore the well-known large cell-cell variability in the progression to senescence, which may blur the sequence of individual cellular events.

Based on pioneering work in yeast (*Ryley and Pereira-Smith, 2006*) and bacteria (*Wang et al., 2010*), the development of microfluidics-based mother cell traps has partially alleviated these limitations by allowing continuous observation of individual cell divisions and relevant fluorescent cellular markers under the microscope from birth to death (*Lee et al., 2012*; *Xie et al., 2012*; *Fehrmann et al., 2013*). In these studies, monitoring individual cells over time in a microfluidic device has demonstrated the unique potential to quantitatively characterize the heterogeneity in cellular dynamics during aging. Recent years have seen a wide diversification of microfluidic devices aimed at improving both experimental throughput and cell retention rates (*Jo et al., 2015*; *Liu et al., 2015*; *Li et al., 2017*). These new developments have helped to highlight the existence of independent trajectories leading to cell death (*Li et al., 2017*; *Morlot et al., 2019*; *Li et al., 2020*) and to better understand the physiopathology of the senescent state (*Neurohr et al., 2018*).

However, the hype surrounding these emerging microfluidic techniques has so far masked a key problem associated with high-throughput time-lapse imaging, namely the difficulty of extracting quantitative information in an efficient and standardized manner due to the manual aspect of the analysis (*Huberts et al., 2014*). In theory, expanding the number of individual cell traps and chambers on a microfluidic system makes it possible to test the effect of a large number of genetic and/or environmental perturbations on aging. Yet, in practice, this is out of reach since lifespan analyses require manual division counting and frequent corrections in cell segmentation. This problem has largely limited the interest of the 'arms race' observed in recent years for the temporal tracking of individual cells during aging. This has also made it very difficult to cross-validate the results obtained by different laboratories, which is yet essential to advance our understanding of the mechanisms involved in aging.

Fortunately, the rapid development of powerful deep learning-based image processing methods in biology using convolutional neural networks (CNN) (*Laine et al., 2021*) suggests a way to overcome this important technical barrier. Recently, a study showed the potential of image classification by a CNN or a capsule network to classify the state of dividing yeast cells (i.e. budded, unbudded, etc.) trapped in a microfluidic device (*Ghafari et al., 2021*). However, due to the limited accuracy of the model, it has not demonstrated its ability to perform an automated division counting, let alone determine the RLS of individual cells. This is likely due to the fact that the misclassification of a single frame during the lifespan can dramatically compromise the accuracy of the RLS measurement.

Here, we report the development of DetecDiv, an integrated platform that combines high-throughput observation of cell divisions using a microfluidic device, a simple benchtop image acquisition system, and a deep learning-based image processing software with several image classification frameworks. Using this methodology, one can accurately track successive cell divisions in an automated manner and reconstruct RLS without human intervention, saving between 2 and 3 orders of magnitude on the analysis time. By combining this pipeline with additional deep-learning models for time-series classification and semantic segmentation, we provide a comprehensive toolset for an in-depth quantification of single-cell trajectories (i.e. division rate, mortality, size, and fluorescence) during entry into senescence and adaptation to environmental stress.

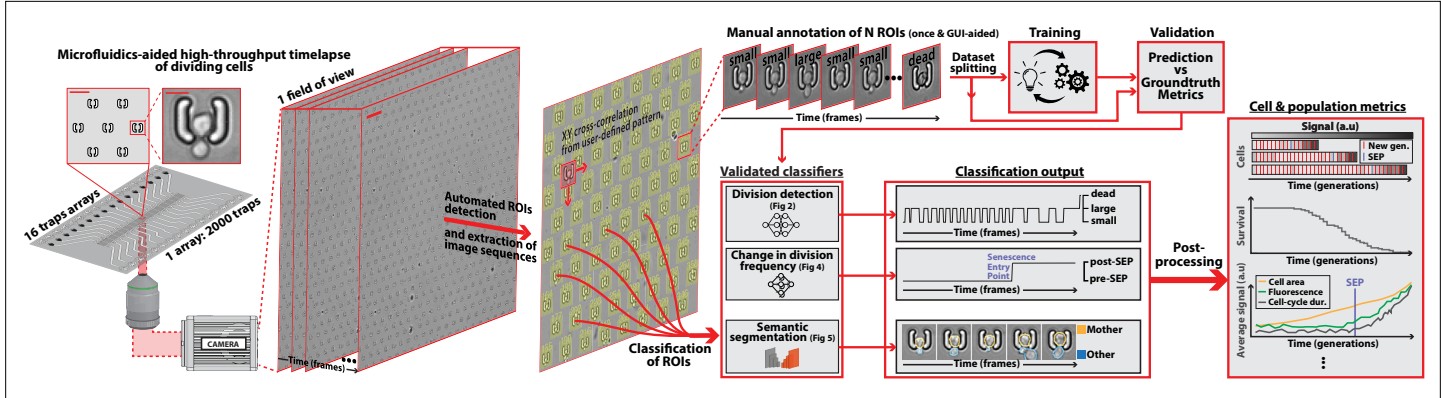

**Figure 1.** DetecDiv workflow Left: Sketch of the analysis pipeline used to track divisions at the single-cell level. Left: A microfluidic device, featuring 16 independent channels with 2000 individual cell traps in each (depicted with a zoom on the trap array (scale bar: 20 μm) and zoom on one trap containing a budding yeast (scale bar: 5 μm)), is imaged using time-lapse microscopy. Middle-left: Typical temporal sequence of brightfield field of views obtained with the setup (scale bar: 60 μm). Regions Of Interest (ROI) representing the traps are automatically detected using XY cross-correlation processing, and the temporal sequence of each ROI (trap) is extracted and saved. Top-right: Sketch of the training and validation pipeline of DetecDiv classifiers. A set of ROIs is picked from one (or several) experiments and annotated to form a groundtruth dataset. It is then split into a training set, used to train the corresponding classifier, and a test set used to validate the trained classifier. Bottom-right: Example of signals extracted from ROIs using DetecDiv classifiers. An image classifier can be used to extract oscillations of classes describing the size of the bud, from dividing cells, and thus the occurrence of new cell cycles (more details in *Figure 2*). A sequence classifier can be used to detect changes in cell-cycle frequency, such as a cell-cycle slowdown (Senescence Entry Point, SEP; more details in *Figure 4*). A pixel classifier can be used to segment the mother cell from other cells, and from the background (more details in *Figure 5*). Using these classifiers on the same ROIs allows extracting quantitative metrics from dividing cells, at the single-cell and population level.

The online version of this article includes the following figure supplement(s) for figure 1:

**Figure supplement 1.** Experimental setup and microfluidic device.

## Results

## Building an improved microfluidic device and a minimal image acquisition system for replicative lifespan analyses

The primary scope of our present study was to overcome the current limitations inherent to the analysis of large-scale replicative lifespan assays by taking advantage of deep-learning image processing methods. Yet, we took this opportunity to provide improvements to individual mother cell trapping devices, in order to maximize the robustness of RLS data acquisition. Based on a design similar to that reported in previous studies (*Jo et al., 2015*; *Crane et al., 2014*; *Liu et al., 2015*), we added small jaws on the top of the trap to better retain the mother cells in the traps (especially the old ones *Figure 1* and *Figure 1—figure supplement 1G*). In addition, we reduced the wall thickness of the traps to facilitate their deformation and thus avoid strong mechanical constraints when the cells become too big (*Figure 1—figure supplement 1D,G* and supplementary text for details). Finally, we added a microfluidic barrier that filters cells coming from microcolonies located upstream of the trap matrix, which eventually clog the device and thus compromise the experiment after typically 24 h of culture. Altogether, the microfluidic device features 16 independent chambers with 2000 traps each, eliciting multiple conditions and strains to be analyzed in parallel.

Next, we built a custom benchtop microscope (referred to as the 'RAMM system' in the following, see methods for details) using simple optical parts to demonstrate that high-throughput division counting and quantitative RLS assays do not require any expensive fully-automated or high-magnification commercial microscopy systems. For this, we used a simple rigid frame with inverted epifluorescence optics, a fixed dual-band GFP/mCherry filter set, a brightfield illumination column, a camera, and a motorized stage, for a total cost of fewer than 40 k euros (*Figure 1—figure supplement 1A-B*). Image acquisition, illumination, and stage control were all interfaced using the open-source Micromanager software (*Edelstein et al., 2014*). Using a ×20 magnification objective, this 'minimal' microscope allowed us to follow the successive divisions and the entry into senescence of typically 30,000 individual cells in parallel with a 5 min resolution (knowing that there are ~500 traps per field of view using the ×20 objective).

## An image sequence classification model for automated division counting and lifespan reconstruction

This image acquisition system generates a large amount of cell division data (on the Terabytes scale depending on the number of channels, frames, and fields of view), only a tiny part of which can be manually curated in a reasonable time. In particular, the determination of replicative lifespans requires counting successive cell divisions until death, hence, reviewing all images acquired for each cell in each field of view over time. In addition, automating the division counting process is complicated by the heterogeneity in cell fate (i.e. cell-cycle durations and cell shape), especially during the entry into senescence.

To overcome this limitation, we have developed an image classification pipeline to count successive generations and reconstruct the entire lifespan of individual cells dividing in the traps (*Figure 2A*). For this, we have trained a convolutional neural network (CNN) based on the 'Inception v1' architecture (*Szegedy et al., 2015*) to predict the budding state of the trapped cells by assigning one of six possible classes (unbudded, small-budded, large-budded, dead, empty trap, and clogged trap) to each frame (*Figure 2A*, Top). In this framework, the alternation between the 'large budded' or 'unbudded' and the 'small budded' states reveals bud emergences. The cell cycle durations can be deduced by measuring the time interval between successive budding events, and the occurrence of the 'dead' class determines the end of the cell's lifespan (*Figure 2A*, Bottom). We selected this classification scheme - namely, the prediction of the budding state of the cell - over the direct assessment of cell division or budding (e.g. 'division' versus 'no division') because division and budding events can only be assessed by comparing successive frames, which is impossible using a classical CNN architecture dedicated to image classification, which takes a single frame as input. To train and evaluate the performance of the classifier, we generated a manually annotated dataset (referred to as 'groundtruth' in the following) by arbitrarily selecting 250 traps (split into a training and a test set, see Methods) containing situations representative of all cellular states from different fields of view and independent experimental replicates.

Benchmarking the classifier consisted of three steps: first, we computed the confusion matrices (*Figure 2—figure supplement 2A*) as well as the classical metrics of precision (i.e. the fraction of correct predictions among all predictions for each class), recall (i.e. the fraction of detected observations among all observations for each class), and $F_1$-score (i.e. the harmonic mean of precision and recall). The $F_1$-score was found to be higher than 85% for all classes (*Figure 2—figure supplement 2C*). Next, the predictions of budding events were compared to the manually annotated data. Despite a good visual match between the groundtruth and the CNN predictions, the distribution of cell-cycle durations revealed that the model tends to predict 'ghost' divisions of abnormally short duration (*Figure 2B*). In addition, sporadic misclassification could falsely assign a cell to the 'dead' state, thus decreasing the number of total generations predicted based on the test dataset (N=1127 for the groundtruth versus N=804 for the CNN model, see *Figure 2C*). Last, by comparing the lifespan predictions to the corresponding groundtruth data, we observed a striking underestimate of the overall survival (*Figure 2D*), due to the sporadic misassignments of the 'dead' class (*Figure 2—figure supplement 1B*).

These problems could be partially alleviated by post-processing the predictions made by the CNN (see 'CNN+PP' in *Figure 2B–D* and supplementary text for details). Indeed, by ignoring isolated frames with a 'dead' class, we could greatly reduce the number of cases with premature cell death prediction, yet we failed to efficiently remove ghost divisions, hence leading to an overestimate of the RLS and a large number of short cell-cycles (*Figure 2C–D*).

An inherent limitation to this approach is that images are individually processed without taking the temporal context into account. Although a more complex post-processing routine could be designed to improve the robustness of the predictions, it would come at the expense of adding more ad hoc parameters, hence decreasing the generality of the method. Therefore, to circumvent this problem, we decided to combine the CNN image classification with a long short-term memory network (LSTM) (*Venugopalan et al., 2015*; *Hochreiter and Schmidhuber, 1997*), to take into account the time-dependencies between images (*Figure 2A*, Middle). In this framework, the CNN was first trained on the individual images taken from the training set similarly as above. Then, the CNN network activations computed from the temporal sequences of images were used as inputs to train an LSTM network (see supplementary text for details). Following this training procedure, the assembled CNN+LSTM

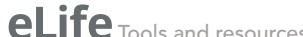

**Figure 2.** DetecDiv cell-cycle duration predictions and RLS reconstruction pipeline. (**A**) Principles of the DetecDiv division tracking and lifespan reconstruction pipeline; Brightfield images are processed by a convolutional neural network (CNN) to extract representative image features. The sequence of image features is then processed by a long short-term memory network (LSTM) that assigns one of the 6 predefined classes ('unbud', 'small', 'large', 'dead', 'clog', 'empty'), taking into account the time dependencies. Temporal oscillations between 'large' and 'small' or 'large' and

*Figure 2 continued on next page*

*Figure 2 continued*

'unbudded' indicate the beginning of a new generation (i.e. cell-cycle). The appearance of the 'dead' class marks the end of the lifespan. For scale reference, each image is 19.5μm wide. (**B**) Comparison of the different methods used for six sample cells. The gray bars represent the groundtruth data made from manually annotated image sequences. Colored lines indicate the corresponding predictions made by CNN+LSTM (orange), the CNN+post-processing (magenta), and the CNN (blue) networks (see Methods and supplementary text for details). The red segments indicate the position of new generation events. (**C**) Left: histogram of cell-cycle durations representing groundtruth data and predictions using different processing pipelines. The p-value indicates the results of a rank-sum test comparing the predictions to the groundtruth for the different pipeline variants. The total number of generations annotated in the groundtruth or detected by the networks is indicated in the legend. Right: Scatter plot in log scale representing the correlation between groundtruth-calculated cell-cycle durations and those predicted by the CNN+LSTM network. $R^2$ represents the coefficient of correlation between the two datasets. Precision and recall are defined in the Methods section. (**D**) Left: cumulative distribution showing the survival of cells as a function of the number of generations (N=50 cells). The numbers in the legend indicate the median replicative lifespans. The p-value indicates the results from a statistical log-rank test. Right: Scatter plot representing the correlation of the replicative lifespans of 50 individual cells obtained from the groundtruth with that predicted by the CNN+LSTM architecture. Inset: same as the main plot, but for the CNN and CNN+Post-Processing pipelines. $R^2$ indicates the coefficient of correlation between the two datasets. (**E**) Replicative lifespans obtained using the CNN+LSTM network for longevity mutants (solid colored lines, genotype indicated). The shading represents the 95% confidence interval calculated using the Greenwood method (*Pokhrel et al., 2008*). The median RLS and the number of cells analyzed are indicated in the legend. The dashed lines with shading represent the hazard rate (i.e. the instantaneous rate of cell mortality in the population of cells at a given replicative age) and its standard deviation estimated with a bootstrap test (N=100). Results from log-rank tests (comparing WT and mutant distributions) are indicated on the left of the legend. (**F**) Same as E but for WT cells grown in 2% glucose or 2% galactose (colored lines). Inset: Same as C - Left but with the same conditions as the main panel.

The online version of this article includes the following video, source data, and figure supplement(s) for figure 2:

**Source data 1.** Parameter values used for training the CNN+LSTM classifier.

**Figure supplement 1.** Principles of division tracking and lifespan reconstruction using a CNN-based image classification.

**Figure supplement 2.** Image classification benchmarks obtained with the CNN and the CNN+LSTM architecture.

**Figure supplement 3.** Example of image classification correctly labeling the state of the mother cell, despite the presence of surrounding cells with potentially different states.

**Figure supplement 4.** Image classification benchmarks, cell-cycle duration and RLS prediction using different CNNs.

**Figure supplement 5.** Validation of RLS and cell-cycle durations predictions, for mutants and galactose conditions.

**Figure 2—video 1.** Comparison of the groundtruth with the CNN+LSTM classifier predictions for the cellular state.
https://elifesciences.org/articles/79519/figures#fig2video1

network was then benchmarked similarly as described above. We obtained only a marginal increase in the classification metrics compared to the CNN network (about 90–95% precision and recall for all classes, see *Figure 2—figure supplement 2A-B*). Yet, strikingly, the quantification of cell-cycle durations and cellular lifespan both revealed considerable improvements in the accuracy: 'ghost' divisions were drastically reduced, and the distribution of cell-cycle duration was indistinguishable from that of the groundtruth (p=0.45, *Figure 2C*), and the difference between the number of generations predicted by the network and the actual number was less than 2% (N=1147 and N=1127, respectively, see left panel on *Figure 2C*). In addition, the Pearson correlation coefficient for groundtruth vs prediction was very high ($R^2$=0.996, see right panel on *Figure 2C*). This indicates that mild classification errors may be buffered and hence do not affect the accuracy in the measurements of cell-cycle durations. Moreover, it suggests that the network was robust enough to ignore the budding status of the daughter cells surrounding the mother cell of interest (*Figure 2—figure supplement 3*). Similarly, the predicted survival curve was almost identical to that computed from the groundtruth (p=0.74, *Figure 2D* and *Figure 2—video 1*) and the corresponding Pearson correlation reached 0.991 (vs 0.8 and 0.1 for the CNN+PP and CNN, respectively). Last, in order to determine if the performances of the classifier could be further improved using a more complex CNN, we did a similar analysis using the inception v3 (*Szegedy et al., 2016*) and the inception-resnet v2 (*Szegedy et al., 2017*) networks. We did not observe any increase in classification accuracy (*Figure 2—figure supplement 4*), while the classification times increased with the CNN complexity (*Figure 2—figure supplement 4J*).

Altogether, these benchmarks indicated that only the combined CNN+LSTM architecture provided the necessary robustness to provide an accurate prediction of individual cellular lifespan based on image sequence classification.

Following its validation, we deployed this model to classify all the ROIs from several fields of view extracted from three independent experiments. We were thus able to obtain a survival curve with N=1880 lifespans in a remarkably short time (*Figure 2E*): less than 3.5 s were necessary to classify

the 1000 images in one lifespan using 8 Tesla K80 GPUs (see Methods for details). This is to be compared with manual annotation of images which takes 5–10 min per cell lifespan depending on the number of generations (i.e. computing is ~100 times faster than manual annotation). Conversely, it would have taken a human being between 7 and 14 days, working 24 h a day, to manually annotate 2000 cells (vs. 2 h for the computer). To further apply the classification model trained on images of wild-type (WT) cells, we measured the large-scale RLS in two classical longevity mutants. Remarkably, we recapitulated the increase (resp. decrease) in longevity observed in the *fob1Δ* (resp. *sir2Δ*) mutant (*Defossez et al., 1999*; *Lin et al., 2000*) and we could compute the related death rate with a high-confidence interval thanks to this large-scale dataset (*Figure 2E*). Model predictions were further evaluated by comparing the predicted replicative lifespans to manually generated test sets for each of the mutants (*Figure 2—figure supplement 5A*). In addition, using glucose and galactose as carbon sources, we performed comparative measurements of cell-cycle durations (N=38,205 events and N=15,495 events for glucose and galactose, see *Figure 2F* inset) and RLS (median = 26 generations, N=1174 events and median=24, N=565 events, for glucose and galactose, respectively, *Figure 2F*). Our results were in line with previous measurements (*Liu et al., 2015*; *Frenk et al., 2017*). To further check the performance of the model, we used an additional manually generated test set obtained with cells growing in galactose to compare to the corresponding predictions (*Figure 2—figure supplement 5B-D*). This evaluation demonstrated that the model initially trained with cells growing under glucose conditions could be successfully applied to data obtained with another carbon source, which is known to affect the cell-cycle duration and the general physiology of the cell.

Altogether, our study shows that our classification pipeline can successfully detect cell divisions, perform lifespan replicative analysis with high throughput, and is robust enough to be employed with different strain backgrounds and under various environmental conditions, even though the training has only been performed on WT data and in glucose conditions.

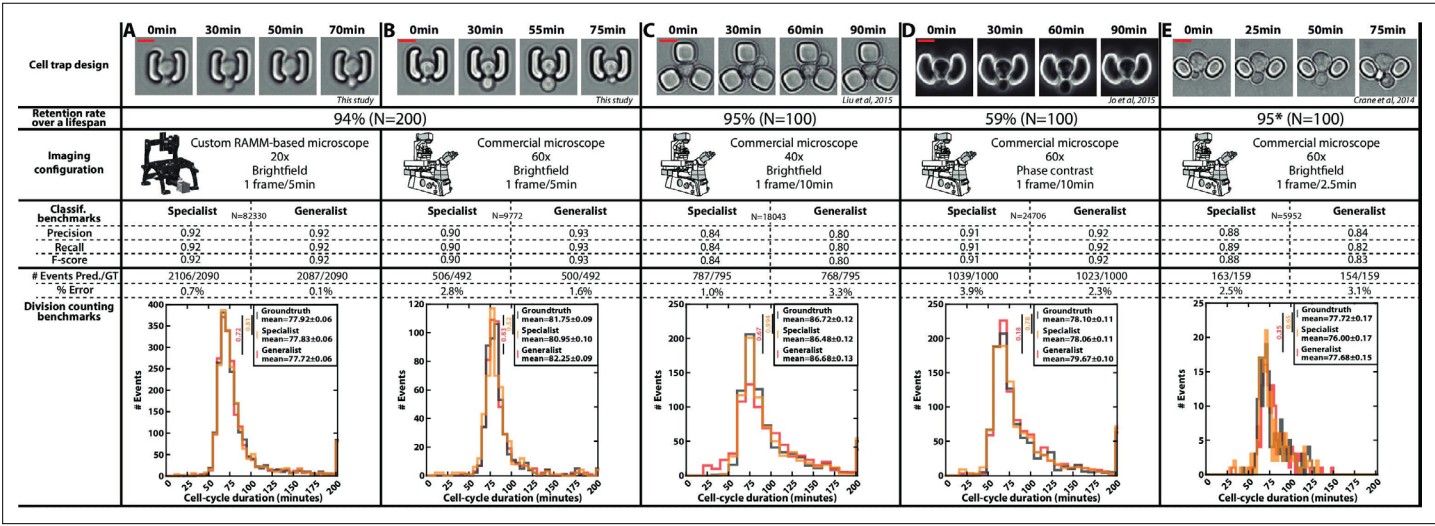

**Figure 3.** Classification benchmarks and performances of the divison detection of a CNN+LSTM image classifier, trained on time-lapses images from different microfluidic devices and imaging setups. A specialist classifier was trained independently for each source, while a generalist classifier was trained on a mixed dataset generated from all the sources. (**A**) Cell trap and imaging setup developed in this study, with a framerate of 1 frame/5 min. (**B**) Cell trap developed in this study imaged with a ×60 objective mounted on a commercial imaging system with a framerate of 1 frame/5 min. (**C**) Cell trap from the Acar lab (*Liu et al., 2015*) imaged with a ×40 objective mounted on a commercial imaging system with a framerate of 1 frame/10 min. (**D**) Cup-shaped trap similar to *Jo et al., 2015*, imaged with a ×60 phase-contrast objective mounted on a commercial imaging system with a framerate of 1 frame/10 min. (**E**) Cell trap from the Swain lab (*Crane et al., 2014*; *Granados et al., 2018*) imaged with a ×60 objective mounted on a commercial imaging system with a framerate of 1 frame/2.5 min. Scale bars: 5µm.

The online version of this article includes the following figure supplement(s) for figure 3:

**Figure supplement 1.** Top: Time-lapse images of traps from the Acar lab (*Liu et al., 2015*) from which the automated division detection can be impaired due to multiple cells in the trap (orange arrows).

## Application of the division counting and lifespan prediction model to different imaging platforms and microfluidic devices

To further test the robustness of our analysis pipeline, we proceeded to the analysis of several datasets obtained under various imaging conditions. First, we performed experiments with the same microfluidic system but using a commercial microscope with ×60 magnification. After training the classifier on 80 ROIs and testing on 40 independent ROIs, we observed similar results to those obtained with the RAMM system and a ×20 objective (compare the 'specialist' columns for the panels in *Figure 3A and B*): the classification benchmarks were greater than 90%, the error rate on the number of generations detected was a few percents, and the cell-cycle length distributions were similar between prediction and groundtruth. This first demonstrated that neither the RAMM imaging system nor the ×20 magnification is required to guarantee successful division counting and lifespan reconstruction with our analysis pipeline.

In addition, we gathered time-lapse microscopy datasets from several laboratories using microfluidic cell trapping systems with different geometries and various imaging conditions (*Figure 3C and E*; *Crane et al., 2014*; *Liu et al., 2015*; *Granados et al., 2018*). We also included data generated in our lab based on a device similar to that used in *Jo et al., 2015* (*Figure 3D*). For each trap geometry, we manually evaluated the retention rate of a mother cell during a lifespan. Indeed, high retention is key to getting a reliable measurement of the RLS (i.e. to ensure that mother cells are not eventually replaced by their daughters). This analysis revealed that a 'semi-open' geometry (as in the design shown in *Figure 3D*, *Figure 1—figure supplement 1G*, and *Figure 3—figure supplement 1*) did not prevent large mother cells from being sporadically expelled during the budding of their daughters, unlike other cell trap shapes. Of note, the geometry proposed by *Crane et al., 2014*; *Figure 3E* was not tested on an entire lifespan, but only on about even generations, hence leading to an overestimation of the retention rate (it was reported to be below 50% in the original paper).

For each dataset, we trained a specific classifier (or 'specialist') on 80 ROIs and validated it on 40 independent ROIs. The different benchmarks (i.e. classification performance and division predictions) showed that each specialist performed very well on each specific test dataset, thus confirming further that our analysis pipeline is robust and applicable to different cell trapping and imaging configurations.

Last, instead of training the classifiers separately on each dataset, we asked whether a unique classifier would have sufficient capacity to handle the pooled datasets with all imaging conditions and trap geometries used in *Figure 3*. Strikingly, this "generalist" model showed comparable performance to the different specialists. This approach thus further highlighted the versatility of our methodology and demonstrated the interest in aggregating data sets to ultimately build a standardized reference model for counting divisions, independently of the specific imaging conditions.

## Automated quantification of cellular physiological decline upon entry into senescence

Aging yeast cells have long been reported to undergo a cell-cycle slowdown when approaching senescence (*Mortimer and Johnson, 1959*), a phenomenon that we have since quantified and referred to as the Senescence Entry Point or SEP (*Fehrmann et al., 2013*). More recently, we have demonstrated that this quite abrupt physiological decline in the cellular lifespan is concomitant with the accumulation of extrachromosomal rDNA circles (ERCs) (*Morlot et al., 2019*), a long described marker of aging in yeast (*Sinclair and Guarente, 1997*). Therefore, precise identification of the turning point from healthy to pathological state (named pre-SEP and post-SEP in the following, respectively) is essential to capture the dynamics of entry into senescence, and even more so since the large cell-cell variability in cell death makes trajectory alignment from cell birth irrelevant (*Fehrmann et al., 2013*; *Morlot et al., 2019*). Yet, the noise in cell-cycle durations, especially beyond the SEP, can make the determination of this transition error-prone if based on a simple analysis (e.g. thresholding) of the cell-cycle durations. Hence, to achieve a reliable determination of the SEP in an automated manner, we sought to develop an additional classification scheme as follows: we trained a simple LSTM sequence-to-sequence classifier to assign a 'pre-SEP' or 'post-SEP' label (before or after the SEP, respectively) to each frame, using the sequence of cellular state probabilities (i.e. the output of the CNN+LSTM image classifier described in *Figure 2A*) as input (*Figure 4A*). The groundtruth was generated by visual inspection using a graphical user interface representing the budding status of a given cell over time. Same as above, we used 200 manually annotated ROIs for the training procedure and reserved

**Figure 4.** Deep learning-based measurement of the dynamics of entry into senescence. (**A**) Sketch depicting the detection of the Senescence Entry Point (SEP). The temporal sequence of classes probabilities (i.e. unbud, small, large, dead) is fed into an LSTM network that predicts the SEP by assigning one of the two predefined classes pre-SEP or post-SEP to each frame. (**B**) Correlogram showing the correlation between the SEP predicted by the LSTM network and the groundtruth data, obtained as previously described (*Fehrmann et al., 2013*). The gray level coded data points indicate the local density of the points using arbitrary units as indicated by the gray level bar. (**C**) Sample trajectories indicating the successive generations of individual cells (red lines) along with the cell-cycle duration (color-coded as indicated). (**D**) Average cell-cycle duration versus generation index after aligning all individual trajectories from the SEP (*Fehrmann et al., 2013*). Each point represents an average over up to 200 cell trajectories. The error bar represents the standard error-on-mean.

The online version of this article includes the following source data and figure supplement(s) for figure 4:

**Source data 1.** Parameter values used for training the SEP detection classifier.

**Figure supplement 1.** Classification benchmarks for the detection of the onset of Senescence Entry using an LSTM sequence-to-sequence classification.

47 additional ones that were never 'seen' by the network to evaluate the predictions. Comparing the predictions to the groundtruth revealed that we could successfully identify the transition to a slow division mode ($R^2$=0.93, see *Figure 4B–C* and *Figure 4—figure supplement 1*). Hence, we could recapitulate the rapid increase in the average cell-cycle durations after aligning individual trajectories from that transition (*Figure 4D*), as described before (*Fehrmann et al., 2013*). These results show that complementary classifiers can be used to process time series output by other classification models, allowing further exploitation of relevant dynamic information, such as the entry into senescence.

## Cell contour determination and fluorescence image quantification by semantic segmentation

Quantifying the dynamics of successive divisions is an indispensable prerequisite for capturing phenomena that span multiple divisions such as replicative aging. However, in order to make the most of the possibilities offered by photonic microscopy, it is necessary to develop complementary

cytometry tools. For this purpose, semantic segmentation based on the classification of pixels has seen a growing interest recently to process biomedical images since the pioneering development of the U-Net architecture (*Ronneberger et al., 2015*). U-Net networks feature an encoding network that extracts meaningful image information and a decoding part that reconstructs a segmented image with a user-defined number of classes (e.g. background, cell, etc.). Recently, the original U-NET architecture has been employed for segmentation in yeast (*Dietler et al., 2020*). More generally, more sophisticated versions have been released allowing the segmentation of objects with low contrast and/or in dense environments, such as Stardist (*Schmidt et al., 2018*) and Cellpose (*Stringer et al., 2021*).

Here, since the complexity of images with individual cell traps is limited, we have used an encoder/ decoder network based on the DeepLabV3 + architecture (*Chen et al., 2018*, *Figure 5—figure supplement 1*), to segment brightfield images (*Figure 5A*, *Figure 5—video 1*, and Methods). Briefly, DeepLabV3+ features an encoder/decoder architecture similar to U-Net, but is more versatile by allowing to process images of arbitrary size. In the following, we chose the Resnet50 network (*He et al., 2016*) as the CNN encoder, which we found to outperform the Inception model for this task. We trained the model on ~1,400 manually segmented brightfield images using three output classes (i.e. 'background,' 'mother cell,' 'other cell') in order to discriminate the mother cell of interest from the surrounding cellular objects. We used a separate test dataset containing ~500 labeled images to evaluate the performance of the classifier (see Methods for details about the generation of the groundtruth data sets). Our results revealed that mother cell contours could be determined accurately with a trained classifier (*Figure 5A–C* and *Figure 5—figure supplement 2A-D*). In addition, we used a cross-validation procedure based on random partitioning of training and test datasets that highlighted the robustness of the classification (*Figure 5—figure supplement 2E*). Overall, this segmentation procedure allowed us to quantify the dynamics of volume increase of the mother cell during replicative aging (*Figure 5C–D*), as previously reported (*Morlot et al., 2019*).

Last, a similar training procedure with ~3000 fluorescence images with a nuclear marker (using a strain carrying a histone-Neongreen fusion) yielded accurate nuclei contours (*Figure 5E–F*, *Figure 5— figure supplement 3*). It successfully recapitulated the sharp burst in nuclear fluorescence that follows the Senescence Entry Point (*Figure 5G–H*; *Morlot et al., 2019*).

## Automated quantitative measurements of the physiological adaptation to hydrogen peroxide

Beyond replicative longevity analyses, we wondered if this automated pipeline could be applied to other biological contexts, in which cell proliferation and cell death need to be accurately quantified over time. Hence, we sought to measure the dynamics of the physiological adaptation of yeast cells subjected to hydrogen peroxide stress.

For this purpose, young cells were abruptly exposed to different stress concentrations, ranging from 0 to 0.8 mM $H_2O_2$, and observed over about 15 h (*Figure 6A*). We used a strain carrying the Tsa1-GFP fusion protein (*TSA1* encodes a peroxiredoxin, a major cytosolic antioxidant overexpressed in response to oxidative stress) as a fluorescent reporter of the cellular response to this stress (*Goulev et al., 2017*).

In this context, we first sought to characterize the dynamics of the cell cycle by using the classifier reported in *Figure 2* - without doing any retraining - to detect divisions during the experiment (using N=250 ROIs). Our automated analysis revealed different possible cell fates, whose proportions varied according to the stress concentration (*Figure 6A and B*): in the absence of stress (0 mM), cells maintained a constant division rate throughout the experiment; in contrast, at 0.3 mM, the population partitioned between cells that recovered a normal division rate after experiencing a delay (see the 'adapted cells' in *Figure 6B*) and others that seemed unable to resume a normal cell-cycle (see the 'slowly dividing cells' in *Figure 6B*), in agreement with previous results (*Goulev et al., 2017*).

Higher doses of stress (0.5 mM and 0.8 mM) saw these populations gradually disappear, indicating that very few divisions occur at these elevated doses. To check this further, we exploited further the outputs of the classifier to score the onset of cell death for each trapped cell (*Figure 6A and C*). Our analysis revealed a progressive, dose-dependent increase in the fraction of dead cells over time, which was confirmed by a comparison between groundtruth data and network predictions of the time of cell death (*Figure 6—figure supplement 1*). These results thus demonstrated the possibility to perform

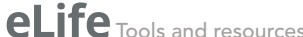

**Figure 5.** Deep learning-based semantic segmentation of cells and nuclei. (**A**) Principles of semantic cell contours segmentation based on brightfield images; Top and middle row: Individual brightfield images were processed by the DeeplabV3 + network that was trained to perform pixel classification using three predefined classes representing the background (black), the mother cell of interest (orange), or any other cell in the image (blue). Bottom row: overlay of brightfield images with segmented cellular contours. For scale reference, each image is 19.5μm wide. (**B**) Correlogram showing the correlation between individual cell area predicted by the segmentation pipeline and the groundtruth data, obtained by manual annotation of the

*Figure 5 continued on next page*

*Figure 5 continued*

images. The color code indicates the local density of the points using arbitrary units. (**C**) Sample trajectories indicating the successive generations of individual cells (red lines) along with the cell surface area (color-coded as indicated). (**D**) Average mother cell surface area versus generation index after aligning all individual trajectories from the SEP (*Fehrmann et al., 2013*). Each point represents an average of up to 200 cell trajectories. The error bar represents the standard error-on-mean. (**E**) Principles of semantic cell nuclei segmentation based on fluorescent images of cells expressing a histone-Neongreen fusion. The semantic segmentation network was trained to classify pixels between two predefined classes ('background' in black, 'nucleus' in green). For scale reference, each image is 19.5μm wide. (**F**) Same as B but for nuclear surface area. (**G**) Same as C but for total nuclear fluorescence (**H**) Same as in D but for total nuclear fluorescence.

The online version of this article includes the following video, source data, and figure supplement(s) for figure 5:

**Source data 1.** Parameter values used for training the classifier dedicated to cell segmentation.

**Source data 2.** Parameter values used for training the classifier dedicated to nucleus segmentation.

**Figure supplement 1.** Principles of the pipeline used for semantic segmentation with DeepLabV3+.

**Figure supplement 2.** Benchmarks for the semantic segmentation of brightfield images.

**Figure supplement 3.** Benchmarks for the semantic segmentation of fluorescence images.

**Figure 5—video 1.** Sample movies of individual cells following cellular state classification, cell, and nuclear contour segmentation.
https://elifesciences.org/articles/79519/figures#fig5video1

real-time and quantitative measurement of the cell death rate in response to an environmental insult, which is rarely precisely done due to the difficulty of precisely scoring dead cells in a population of cells without any additional viability marker.

Finally, we used our semantic segmentation model (reported in *Figure 5*) to quantify cytoplasmic fluorescence over time in the stress response experiment (*Figure 6A*). The population-averaged mean cytoplasmic fluorescence revealed a significant increase at 0.3 mM $H_2O_2$ due to the transcriptional upregulation of antioxidant genes, as previously described (*Goulev et al., 2017*). However, this average upregulation of Tsa1 was lessened at higher doses, an effect we attribute to the large fraction of cell death observed in these conditions (e.g.: bottom cell in *Figure 6A*). Altogether, these results indicate that DetecDiv used with single-cell traps provides a highly suitable method for quantifying both cell division rate and mortality in real-time under variable environmental conditions.

## Discussion

In this study, we have developed a pipeline based on the combined use of two architectures, namely a CNN+LSTM network for the exploitation of temporal information and semantic segmentation (Deep-LabV3+) for the quantification of spatial information. We demonstrate that it can successfully characterize the dynamics of multi-generational phenomena, using the example of the entry into replicative senescence in yeast, a difficult case study that has long resisted any automated analysis. We also successfully used our classification model to score cellular adaptation and mortality in the context of the physiological stress response to hydrogen peroxide. Furthermore, we have developed a graphical user interface to make this method accessible for the community without requiring any programming knowledge. We envision that this methodology will unleash the potential of microfluidic cell trapping devices to quantify temporal physiological metrics in a high-throughput and single-cell manner.

The major novelty of this work lies in the development of an analysis method to automatically obtain survival curves and cytometric measurements during the entry into senescence from raw image sequences. Nevertheless, we also focused our efforts on improving traps to increase the efficiency of RLS assays in microfluidics. Also, we have built a minimal optical system (yet with a motorized stage) assembled from simple optical components (i.e. no filter wheel, fixed objective), for a price of about one-third that of a commercial automated microscope, which can be made accessible to a larger community of researchers. Although many laboratories use common imaging platforms with shared equipment, it is important to note that the cost of an hour of microscopy is around 10–20 euros in an imaging facility. As an RLS assay typically lasts 60–80 hours, these experiments may not be affordable. Developing a simple system can therefore quickly pay off if the lab does not have its own microscope.

Using this experimental setup, we showed that our analysis pipeline works perfectly even with a low optical resolution (i.e. the theoretical resolution of our imaging system with a ×20, N.A. 0.45 objective is ~0.7 μm), and without any contrast-enhancing method. In practice, it might be desirable

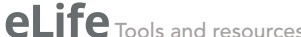

**Figure 6.** Automated analysis of the stress response to $H_2O_2$ using DetecDiv. (**A**) Successive brightfield and Tsa1-GFP images of three representative cells submitted to 0.3 mM of $H_2O_2$ and corresponding to a different fate. The orange contour of the cell is determined using the segmentation described in *Figure 5*, and the total GFP fluorescence inside it is depicted as a function of time, where red bars indicates a new generation and the purple dotted bar indicated the onset of $H_2O_2$. For scale reference, each image is 19.5μm wide. (**B**) Scatter plot of automatically detected cell-cycle

*Figure 6 continued on next page*

*Figure 6 continued*

durations versus time of 500 cells submitted to different doses of $H_2O_2$. The purple area indicates the presence of the indicated dose of $H_2O_2$. (**C**) Fraction of dead cells versus time as automatically detected by the CNN+LSTM classifier, under different $H_2O_2$ doses. The purple area indicates the presence of the indicated dose of $H_2O_2$. N=500. (**D**) Mean Tsa1-GFP fluorescence from cells submitted to different doses of $H_2O_2$. The purple area indicates the presence of the indicated dose of $H_2O_2$. N=500.

The online version of this article includes the following figure supplement(s) for figure 6:

**Figure supplement 1.** Validation of predicted time of death and death status at the end of the experiment.

for some applications to use higher magnification to better preserve spatial information and analyze the detailed localization of fluorescent markers. Yet, using the same microfluidic device described here, we showed that DetecDiv works similarly with higher magnification objectives and different imaging systems. In addition, we demonstrated that division detection can also be performed with cells growing in traps with different geometries (*Figure 3*, *Figure 1—figure supplement 1G*, and *Figure 3—figure supplement 1*). Furthermore, a unique classifier trained on a large collection of images obtained under broad imaging contexts can also achieve accurate division detection. This may be instrumental to standardize the quantitative analysis of replicative lifespan data in the yeast aging community.

However, one limitation to applying our analysis pipeline with a broad range of trap geometries is that the accuracy of RLS measurements may be affected when using designs with a low retention rate. Although lifespan trajectories can be marked as 'censored' when the mother cell leaves the traps (as proposed in a recently released preprint *Thayer et al., 2022*), our method is currently unable to systematically detect when a mother cell is replaced by its daughter (e.g. cell traps in *Figure 3D*). Therefore, we believe that retention is an essential feature to consider when designing the geometry of a trap.

An important advantage of individual cell trapping is that it makes image analysis much simpler than using chambers filled with two-dimensional cell microcolonies. Indeed, individual traps behave as a 'hardware-based cell tracking' method, thus alleviating the need to identify and track objects spatially, a procedure that provides an additional source of errors. Because the cells of interest are located in the middle of the traps, the learning process can focus the attention of the classifier on the state of the mother cell only (e.g. small-budded, large-budded, etc.), hence the specific state of the few cells surrounding it may not influence the reliability of the classification of the mother (*Figure 2—figure supplement 3* for specific examples). In addition, a powerful feature of whole image classification is that it can easily be coupled to a recurrent neural network (such as an LSTM network), thus opening the door to more accurate analyses that exploit temporal dependencies between images, as demonstrated in our study.

Beyond the tracking of successive divisions, complementary methods are necessary to characterize the evolution of cell physiology over time. In our study, we used semantic segmentation to delineate the contours of cell bodies over time. Same as above, the ability to discriminate the mother cell of interest from the surrounding cells results is facilitated by the conserved position of the mother at the center of the trap. However, a limitation of our classification scheme is that the buds that arise from the mother cell can not be identified, and further work is necessary to assess the requirements (e.g. the size of the training set) to achieve their successful segmentation, such as using a separate 'bud' class. Thus, it is currently impossible to measure the growth rate (in volume) of the mother cell over time (most of the biomass created during the budded period goes into the bud) and it precludes analyzing fluorescent markers that would localize into the bud. Future work may explore how the use of additional segmentation classes or the use of tracking methods could complement our pipeline to alleviate this limitation, as recently shown (*Pietsch et al., 2022*). Alternatively, the development of an instance segmentation method (*He et al., 2017*; *Prangemeier et al., 2022*) could also facilitate the identification and separation of different cell bodies in the image.

Unlike classical image analysis methods, which require complex parameterization and are highly dependent on the problem being addressed, the well-known advantage of machine learning is the versatility of the models, which can be used for a variety of tasks. Here, we show that our division counting/lifespan reconstruction classifier can readily be used to quantify cellular dynamics and mortality in response to hydrogen peroxide stress. We envision that DetecDiv could be further applied in different contexts without additional development - yet with potential retraining of the classifier with

complementary data, and/or following the definition of new classes. For example, it could be useful to develop a classifier able to identify different cell fates during aging based on image sequences (e.g. *petite* cells (*Fehrmann et al., 2013*), or mode 1 versus mode 2 aging trajectories *Jin et al., 2019*), as well as during induced (*Bagamery et al., 2020*) or undergone (*Jacquel et al., 2021*) metabolic changes. More generally, the rationalization of division rate measurements in a system where there is no competition between cells offers a unique framework to study the heterogeneity of cell behaviors in response to environmental changes (stress, chemical drugs, etc.), as demonstrated in our study and evidenced by the rise of high-throughput quantitative studies in bacteria (*Bakshi et al., 2021*). Mechanistic studies of the cell-cycle could also benefit from a precise and standardized phenotypic characterization of the division dynamics. Along this line, beyond the classification models described in this study, we have integrated additional frameworks, such as image and image sequence regressions (see *Supplementary file 2* for details), which could be useful to score fluorescent markers quantitatively and over time (e.g. mitotic spindle length inference, scoring of the mitochondrial network shape, etc.). We envision that the kind of approach described here may be easily transferred to other cellular models to characterize heterogeneous and complex temporal patterns in biological signals.

## Materials and methods

### Strains

All strains used in this study are congenic to S288C (see *Supplementary file 1* for the list of strains). See the next section for detailed protocols for cell culture.

### Cell culture

For each experiment, freshly thawed cells were grown overnight, diluted in the morning, and allowed to perform several divisions (~5 hours at 30 °C) before injection into the microfluidic device. Yeast extract Peptone Dextrose (YPD) medium was used throughout the experiments.

### Microfabrication

The designs were created on AutoCAD to produce chrome photomasks (jd-photodata, UK). The microfluidic master molds were then made by standard photolithography processes. The designs were created on AutoCAD (see https://github.com/TAspert/DetecDiv_Data; *Aspert, 2021* to download the design) to produce chrome photomasks (jd-photodata, UK). Then, the microfluidic master molds were made using two rounds of classical photolithography steps.

The array of 2000 traps was created from a 5.25 μm deposit by spinning (WS650 spin coater, Laurell, France) 3 mL of SU8-2005 negative photoresist at 2500 rpm for 30 s on a 3″ wafer (Neyco, FRANCE). Then, a soft bake of 3 min at 95 °C on heating plates (VWR) was performed, followed by exposure to 365 nm UVs at 120 mJ/cm² with a mask aligner (UV-KUB3, Kloé, FRANCE). Finally, a post-exposure bake identical to the soft bake was performed before development using SU-8 developer (Microchem, USA).

The second layer with channel motifs was made of a 30 μm deposit of SU8-2025, by spinning it at 2500 rpm for 30 s. Subsequently, a soft bake of 3 min at 65 °C and 6 min at 95 °C was performed. The wafer was then aligned with the mask containing the motif of the second layer before a 120mJ/cm² exposure. A post-exposure bake similar to the soft bake was then performed.

After each layer, we performed a hard bake at 150 °C for 15 min to anneal potential cracks and stabilize the photoresist. Finally, the master molds were treated with chlorotrimethylsilane to passivate the surface.

### Microfluidics chip design, fabrication, and handling

The microfluidic device is composed of an array of 2048 microstructures able to trap a mother cell while removing successive daughter cells, similarly to previously designed (*Ryley and Pereira-Smith, 2006*; *Zhang et al., 2012*; *Lee et al., 2012*; *Crane et al., 2014*; *Jo et al., 2015*; *Liu et al., 2015*). The traps are composed of two symmetrical structures separated by 3 μm (see *Figure 1* and *Figure 1—figure supplement 1D-G*), in such a way that only one cell can be trapped and remain in between the structures. We have measured that 94% of the cells that underwent at least five

divisions in the trap would stay inside until their death. This is striking contrast with the results obtained with a device with a semi-open trap geometry (see *Figure 3* and *Figure 3—figure supplement 1* for details). Moreover, a particle filter with a cutoff size of 15 µm is present before each array of traps, preventing dust particles or debris from clogging the chip (*Figure 1—figure supplement 1C*). In addition, a cell filter with a cutoff size of 1.5µm is placed upstream of each trapping area to prevent contamination of the inlet by cells during the seeding phase (see next paragraph).

The microfluidic devices were fabricated using soft-lithography by pouring polydimethylsiloxane (PDMS, Sylgard 184, Dow Chemical, USA) with its curing agent (10:1 mixing ratio) on the different molds. The chips were punched with a 1 mm biopsy tool (KAI, Japan) and covalently bound to a 24×50 mm coverslip using plasma surface activation (Diener Zepto, Germany). The assembled chips were baked for 30 min at 60 °C to consolidate covalent bonds between glass and PDMS. The chip was then plugged using a 1 mm Outer Diameter (O.D.) PTFE tubing (Adtech, UK) and the channels were primed using culture media for 5 min. After that, cells were injected through the outlet using a 5 mL syringe and a 26 G needle for approximately 1 min per channel by applying very gentle pressure. The cell filter placed upstream of the trapping area prevented the cells from entering the tubing connected to the inlet (*Figure 1—figure supplement 1 E*). Then, the inlet of each microfluidic channel was connected to a peristaltic pump (Ismatec, Switzerland) with a 5 µL/min rate to ensure a constant replenishment of the media and dissection of the daughter cells (*Figure 1—figure supplement 1E*). This procedure avoids potential contamination by cells forming colonies upstream of the trapping area, which would induce the clogging of the device after 1–2 days of experiment, therefore making long-lasting experiments more robust.

## Microscopy

The microscope was built from a modular microscope system with a motorized stage (ASI, USA, see the supplementary text for the detailed list of components), a ×20 objective 0.45 (Nikon, Japan) lens, and an sCMOS camera (ORCA flash 4.0, Hamamatsu, Japan). A dual-band filter (#59022, Chroma Technology, Germany) coupled with a two-channel LED system (DC4104 and LED4D067, Thorlabs, USA). The sample temperature was maintained at 30 °C thanks to a heating system based on an Indium Thin Oxide coated glass and an infrared sensor coupled to an Arduino-based regulatory loop.

### Microscope

The microscope was built from a modular microscope system (RAMM, ASI, USA) with trans- (Oly-Trans-Illum, ASI, USA) and epi- (Mim-Excite-Cond20N-K, ASI, USA) illumination. This microscope frame provides a cost-effective solution to build a minimal microscopy apparatus to perform robust image acquisition over several days (*Figure 1—figure supplement 1B*).

It is equipped with a motorized XY stage (S551-2201B, ASI, USA), a stage controller (MS200, ASI, USA), and a stepper motor to drive the ×20 N.A. 0.45 Plan Fluor objective (Nikon, Japan) and an sCMOS camera (ORCA flash 4.0, Hamamatsu, Japan) with 2048 pixels × 2048 pixels (i.e. 650 µm × 650 µm field of view at ×20 magnification). We used a dual-band filter (#59022, Chroma Technology, Germany) coupled with two-channel LED illumination (DC4104 and LED4D067, Thorlabs, USA), which allows fast imaging of GFP and mCherry without any filter switching.

### Sample holder and temperature control

We designed a custom 3D-printed sample holder (by extruding PLA material with a MK3S+printer, Prusa Research, Czech republic) for the microfluidic device to ensure the mechanical stability of the microfluidic device (design available on github: https://github.com/TAspert/ITO_heating_device; *Aspert, 2022*).

In addition, we developed a custom temperature control system to maintain a constant temperature (30 °C) and guarantee optimal cell growth throughout the experiment. Briefly, we used an Indium Tin Oxide (ITO) Coated glass, an electrically conductive and transparent material, in direct contact with the PDMS chip. Therefore, applying a voltage to the ITO glass heat the glass and the adjacent PDMS chip by Joule effect (*Figure 1—figure supplement 1A*).

To achieve a temperature control loop, we used an infrared sensor attached to the objective and facing towards the bottom glass coverslip in contact with the cells. The sensor allowed in situ

temperature measurement and was used in an Arduino-based PID control loop to regulate the heating power to maintain the setpoint temperature (the circuit diagram is available on github: https://github.com/TAspert/ITO_heating_device; *Aspert, 2022*). About 0.3 W was sufficient to maintain a 30 °C temperature at room temperature. Notably, the temperature profile obtained with this method was homogenous and constant throughout the experiment. Furthermore, the glass is fully transparent to visible light. In addition, it does not interfere with fluorescent light when using an inverted microscope since it is located at the end of the optical path, after the sample.

## Software and time-lapse acquisition parameters

Micromanager v2.0. was used to drive the camera, the light source, the XYZ controller, and the LED light source for fluorescence epi-illumination. We developed a specific program in order to drive the temperature controller from the Arduino (The source code is available on github: https://github.com/TAspert/ITO_heating_device; *Aspert, 2022*).

Unless specified otherwise, the interval between two brightfield frames for all the experiments was 5 min, and images were recorded over 1000 frames (i.e. ~3 days). We used three z-stack for brightfield imaging (spaced by 1.35 µm) to ease the detection of small buds during the image classification process for cell state determination. Fluorescent images were acquired with a 10 min interval using 470 nm illumination for 50ms. Up to 80 fields of view were recorded over the 5 min interval.

## Autofocusing

To keep a stable focus through the whole experiment, we developed a custom software-based autofocus routine that finds the sharpest image on the first field of view and then applies the focus correction to the rest of the positions (https://github.com/TAspert/DetecDiv_Data; *Aspert, 2021*). This method provides faster scanning of all fields of view in a reasonable time. Nevertheless, it is almost as efficient as performing autofocusing on each position since the primary source of defocusing in our setup is the thermal drift, which applies identically to all the positions.

## Time-lapse routine

Micromanager v2.0 (*Edelstein et al., 2014*) was used to drive all hardware, including the camera, the light sources, and the stage and objective motors. We could image approximately 80 fields of view (0.65mm×0.65mm) in brightfield and fluorescence (using a dual-band GFP-mCherry filter) with this interval. In the $H_2O_2$ stress response experiments, cells were exposed abruptly to a medium containing the desired concentration (from 0.3mM to 0.8 mM) and fluorescence was acquired every 15 min.

## Additional datasets for the comparative study of division detection

Time-lapse image datasets of individual mother cells trapped in microfluidic devices were obtained from the Murat Acar and Peter Swain lab. The datasets were used to compare the performance of our cell division tracking pipeline, as described in the main text. Data from the Acar lab were generated on a Nikon Ti Eclipse using ×40 brightfield imaging with a 10min-interval and a single z-stack, as previously described (*Liu et al., 2015*). Data from the Swain lab were obtained using a Nikon Ti Eclipse microscope using ×60 brightfield imaging, a 2.5min-interval (*Crane et al., 2014*; *Granados et al., 2018*), and 5 z-stacks combined into a single RGB image and used as input to the classifier. We also used a separate trap design from our own lab that is similar to a previously reported design (*Jo et al., 2015*) which was imaged on a Nikon Ti Eclipse microscope using a ×60 phase-contrast objective.

## Image processing

### DetecDiv software

We developed Matlab software with a graphical user interface, DetecDiv, which provides different classification models: image classification, image sequence classification, time series classification, and pixel classification (semantic segmentation), see *Supplementary file 2* for details. DetecDiv was developed using Matlab, and additional toolboxes (TB), such as the Computer Vision TB, the Deep-learning TB, and the Image Processing TB. A graphical user interface (GUI) was designed to facilitate the generation of the training sets. The DetecDiv software is available for download on GitHub: https://github.com/gcharvin/DetecDiv (*Aspert, 2021*).

DetecDiv can be used with an arbitrarily large number of classes, image channels, types, and sizes. Indeed, several classification models can be defined to process the images: (1) Image classification using a convolutional network (CNN, see *Figure 2—figure supplement 1*); (2) a combined CNN+LSTM classifier (*Figure 2A*); (3) an LSTM network to perform sequence-to-sequence (as in *Figure 4A*) or sequence-to-one classification; (4) An encoder/decoder classifier to perform pixel classification (semantic segmentation) based on the DeeplabV3 + architecture (*Chen et al., 2018*), (*Figure 5*); (5) Similar routines as in 1–4, but for regression analyses (not used in the present study, see *Supplementary file 2* for more details). DetecDiv allows the user to choose among several CNNs -such as GoogleNet or Resnet50 for all image classifications/segmentations applications.

DetecDiv also provides a graphical user interface to generate the groundtruth required for both training and testing the classifiers used in the image classification, pixel classification, and time-series classification pipelines. Furthermore, we paid attention to making this step as user-friendly as possible. For instance, we used keyboard shortcuts to assign labels to individual frames (it takes about 5–10 min to annotate 1000 frames in the case of cell state assignment). Similarly, direct 'painting' of objects with a mouse or a graph pad can be used to label images before launching the training procedure for pixel classification.

DetecDiv training and validation procedures are run either at the command line, which allows using remote computing resources, such as a CPU/GPU cluster, or using a Matlab GUI application. All the relevant training parameters can be easily defined by the user. Moreover, we designed generic routines to benchmark the trained classifiers that allow an in-depth evaluation of the classifiers' performances (*Laine et al., 2021*). Trained classifiers can be exported to user-defined repositories and classified data can be further processed using custom Matlab scripts, and images sequences can be exported as.mat or.avi video files.

Last, DetecDiv provides additional post-processing routines to extract cell-cycle, lifespan and pixel-related (volume, signal intensity, etc) data for further analysis, as performed in the present study.

## Convolutional Neural Networks (CNN) for classification of the cellular budding status and death

We used an image classifier to assess the state of cells in the cell cycle (small, large-budded, etc.) using brightfield images of individual traps. For each frame, we combined the three z-stack images described above into a single RGB image, which was used as input for the classifier.

We defined six classes, four of which represent the state of the cell, that is, unbudded ('unbud'), small-budded ('small'), large-budded ('large'), dead ('dead'), as shown in *Figure 2*. Two additional classes are related to the state of the trap: trap with no cell ('empty') and clogged trap ('clog').

The small class was attributed to images on which the mother cell displayed a bud below a certain threshold size. This size was defined to represent approximately half of the cell-cycle images, but also to stay below the smallest daughter cells.

The large class was attributed to images on which the mother cell displayed a bud above the aforementioned threshold size. If no bud was visible on the mother's surface after the cytokinesis, the image was labeled as "large" if the last daughter cell remained in contact with the mother, or "unbudded" if the mother cell was left alone in the trap.

An image was labeled as 'dead' when the mother cells appeared as dead. This includes an unambiguous, very abrupt (within one frame), and strong change in contrast and appearance of the cell.

The empty class was attributed to images on which no cell was present in the ~three first quarters (from the bottom) part of the trap.

Finally, the clogged class was used for images where more than ~50% of the outer space of the trap was filled with cells.

We trained a pre-trained Inception v1 (also known as GoogleNet) convolutional neural network (*Szegedy et al., 2015*) to classify images according to these six classes using a training set of 200,000 representative manually annotated brightfield images (i.e. 200 ROIs monitored during 1000 frames). Of note, the number of images used for training results from a trade-off between the desired accuracy of the model and the time required to build the training set.

The training of the classifier was achieved using Adaptive Moment estimation (Adam) optimizer (*Kingma and Ba, 2015*). Specific parameters used can be found in the *Figure 2—source data 1*.

After the training procedure, we tested the classifier using a dataset composed of 50 independent ROIs (i.e. ~50,000 images) that were manually annotated and used for benchmarking (*Figure 2— figure supplement 2*).

### Cell-cycle duration measurements and replicative lifespan (RLS) reconstruction based on classification results

As the image classifier outputs a label for each frame corresponding to one of the ix classes defined above, we used the sequence of labels to reveal the successive generations of the cells: the oscillations between the 'large' and 'small' or 'unbudded' and 'small' classes captured the entry into a new cell cycle (i.e. a budding event).

The first occurrence of one of the four following rules was used as a condition to stop the lifespan: (1) the occurrence of a 'Dead' class; (2) division arrest for more than 10 h; (3) occurrence of a 'Clogged' class; (4) the occurrence of an 'Empty' class (the mother left the trap *Figure 1—figure supplement 1G* and *Figure 3—figure supplement 1*), which was a rare case. Premature lifespan arrests due to clogging or mother cell removal (3 and 4) were not considered further for lifespan analyses.

This set of rules was used to compute the cell-cycle duration and the RLS of each individual cell when using either the CNN or the combined CNN+LSTM architecture (see below). However, in order to improve the accuracy of the method based only on the CNN, we implemented an additional 'post-processing' step (referred to as PP in *Figure 2*), namely that two consecutive frames with a 'dead' label are necessary to consider a cell as dead.

### Image sequence classification using combined CNN and a long short-term memory network (LSTM)

To provide a more accurate classification of the image according to the cellular state, we added a bidirectional long short-term memory (LSTM) network with 150 hidden units to the CNN network (*Hochreiter and Schmidhuber, 1997*). The LSTM network takes the whole sequence of images as input (instead of independent images in the case of the CNN). 200 ROIs (with 1000 frames each) were used to train the LSTM network independently of the CNN network (see training parameters in *Figure 2—source data 1* and benchmarks on *Figure 2—figure supplement 2B-D*). The CNN and the LSTM network were then assembled as described in *Figure 2A* in order to output a sequence of labels for each sequence of images. The assembled network was then benchmarked using a set of 50 independent annotated ROIs, as described above.

The training and test datasets are available at: doi.org/10.5281/zenodo.6078462.
The trained network is available at: doi.org/10.5281/zenodo.5553862.

### Assessment of cell-cycle slowdown using an LSTM network

We designed a time series classification method to identify when the cell cycle starts to slow down (S̲enescence E̲ntry P̲oint, or SEP, see Results section). For this, we trained a bidirectional LSTM network with 150 hidden units to classify all the frames in each lifespan, into two classes, 'pre-SEP and 'post-SEP', using a manually annotated dataset containing 200 ROIs. To achieve these annotations, we designed a custom annotation GUI allowing us to monitor the successive states of a mother cell of interest over time, as output by the CNN+LSTM network above. This tool was convenient to detect the cell cycle slow down occurring upon entry into senescence. Then, the LSTM network was trained using class probabilities from the previously described CNN+LSTM (unbudded, small, large, dead), see *Figure 4—figure supplement 1* for benchmarking results on a test set with 47 ROIs.

The training and test datasets are available at: doi.org/10.5281/zenodo.6075691.
The trained network is available at: doi.org/10.5281/zenodo.5553829.

## Brightfield and fluorescence images semantic segmentation using DeepLabV3+

Cells and nuclei contours were determined based on brightfield and fluorescence images, respectively, using the deep learning-based semantic segmentation architecture DeepLabV3+ (*Chen et al., 2018*). To generate the groundtruth data required to feed both the training and test datasets, we developed a graphical user-interfaced routine to 'paint' the input images, a process which took about

15–30 s per image depending on the number of cells in a 60×60 pixel-large field of view. We trained the network using a training set containing 1400 and 3000 images for brightfield and fluorescence images, respectively (see *Figure 5—figure supplement 2* and *Figure 5—figure supplement 3* for benchmarking results). Specific parameters used can be found in the *Figure 5—source data 1* (cell segmentation) and *Figure 5—source data 2* (nucleus segmentation). In addition, we have implemented a cross-validation routine to test the sensitivity of the classifier used for cell segmentation to the training and test datasets. For this purpose, we have performed 30 successive random draws of 200 annotated ROIs to be used as a training set and 50 annotated ROIs for testing the classifier (the total number of manually annotated ROIs is 250). For each draw, we have measured the performance of the classifier (i.e. precision, recall, $F_1$-score), see *Figure 5—figure supplement 2E*.

The training and test datasets are available at: doi.org/10.5281/zenodo.6077125.
The trained network is available at: doi.org/10.5281/zenodo.5553851.

## Classifier benchmarking

We used standard benchmarking to estimate the efficiency of image and pixel classifiers. For each classifier, we computed the confusion matrix obtained by comparing the groundtruth of manually annotated images (or time series) taken from a test set unseen by the network during training to the predictions made by the classifier. We computed the precision, recall, and $F_1$-score for each class (see the corresponding definitions in the Results section). In the specific case of pixel classification (semantic segmentation), we computed these benchmarks for different values of prediction thresholds used to assign the 'mother' and 'nucleus' classes, as reported in *Figure 5—figure supplements 2 and 3*. Then, we performed the segmentation of images using the threshold value that maximizes the $F_1$-score (0.9 and 0.35 for brightfield and fluorescence image classification, respectively).

To benchmark the detection of new generations, we used a custom pairing algorithm to detect false positive and false negative new generation events. Using this, we could compute the precision and recall of the detection of new generation events, and plot the correlation between paired new generations events (*Figure 2D*).

## Statistics

All experiments have been replicated at least twice. Error bars and ± represent the standard error-on-mean unless specified otherwise. Results of specific statistical tests are indicated in the figure legends.

## Computing time

Image processing was performed on a computing server with 8 Intel Xeon E5-2620 processors and 8 co-processing GPU units (Nvidia Tesla K80, released in 2014), each of them with 12Go RAM. Under these conditions, the classification of the time series of 1000 frames from a single trap (roughly 60 pixels × 60 pixels) took 3.5 s to the CNN+LSTM classifier. For the image segmentation, the Deep-LabV3 +network took about 20 s to classify 1000 images. Alternatively, we have used a personal notebook with an Nvidia Quadro T2000 card (similar to a GTX 1650, released in 2019) to set up and troubleshoot the code. Under the conditions, training procedures could be achieved within a few hours or overnight depending on the size of the training set, and classification times were similar to those obtained with a Telsa K80. However, using bigger CNNs, such as the inception v3 or the inception-resnet v2, lead to a large increase in computing times (see *Figure 2—figure supplement 4J*), which makes them cumbersome when using limited computing resources, as in our study.

## Materials availability statement

### Datasets

Annotated datasets and trained classifiers used in this study are available for download as indicated:

- Lifespan analyses:
  - Training and test datasets: doi.org/10.5281/zenodo.6078462
  - Trained Network (CNN+LSTM): doi.org/10.5281/zenodo.5553862
- Brightfield image segmentation:
  - Training and test datasets: doi.org/10.5281/zenodo.6077125
  - Trained Network (Encoder-Decoder DeeplabV3+): doi.org/10.5281/zenodo.5553851
- Cell-cycle slowdown (Senescence Entry Point) detection:

○ Training & test datasets: doi.org/10.5281/zenodo.6075691
○ Trained Network (LSTM): doi.org/10.5281/zenodo.5553829

Information regarding the design of the microfluidic device and of the custom imaging system are available on https://github.com/TAspert/DetecDiv_Data, (copy archived at swh:1:rev:ab-95660be5e0677dba69247d27492036c33e08c1; *Aspert, 2021*).

## Code

The custom MATLAB software DetecDiv, used to analyze imaging data with deep-learning algorithms, is available on https://github.com/gcharvin/DetecDiv.

This software distribution features a tutorial on how to use the graphical user interface: https://github.com/gcharvin/DetecDiv/blob/master/Tutorial/GUI_tutorial.md.

It also provides the main commands to use the DetecDiv pipeline in custom user-defined scripts: https://github.com/gcharvin/DetecDiv/blob/master/Tutorial/commandline_tutorial.md.

A demo project that contains all the necessary files to learn how to use DetecDiv can be downloaded from zenodo: https://doi.org/10.5281/zenodo.5771536.

## Acknowledgements

We thank Audrey Matifas for constant technical support throughout this work, Sophie Quintin and Nacho Molina for carefully reading the manuscript. We are very grateful to Murat Acar and David Moreno Fortuño, as well as Peter Swain and Julian Pietsch for providing the additional time-lapse datasets analyzed in this study. We thank Olivier Tassy for their insightful discussions. We thank Denis Fumagalli at the IGBMC Mediaprep facility for media preparation. We are grateful to the IT service for efficient support and providing the computing resources. We thank the Charvin lab members, Bertrand Vernay, Jerome Mutterer, Serge Taubert, and the IGBMC imaging facility for discussions and technical support. This work was supported by the Agence Nationale pour la Recherche (T.A. and G.C.), the grant ANR-10-LABX-0030-INRT, a French State fund managed by the Agence Nationale de la Recherche under the frame program Investissements d'Avenir ANR-10-IDEX-0002–02.

## Additional information

### Funding

| Funder | Grant reference number | Author |
|---|---|---|
| Agence Nationale de la Recherche | ANR-10-LABX-0030-INRT | Gilles Charvin |
| Agence Nationale de la Recherche | ANR-10-IDEX-0002-02 | Gilles Charvin |

The funders had no role in study design, data collection and interpretation, or the decision to submit the work for publication.

### Author contributions

Théo Aspert, Conceptualization, Data curation, Software, Formal analysis, Validation, Investigation, Visualization, Methodology, Writing – original draft, Writing – review and editing; Didier Hentsch, Resources, Investigation, Methodology; Gilles Charvin, Conceptualization, Data curation, Software, Formal analysis, Supervision, Funding acquisition, Validation, Investigation, Visualization, Methodology, Writing – original draft, Project administration, Writing – review and editing

### Author ORCIDs

Théo Aspert ⬤ http://orcid.org/0000-0003-2957-0683
Gilles Charvin ⬤ http://orcid.org/0000-0002-6852-6952

### Decision letter and Author response
Decision letter https://doi.org/10.7554/eLife.79519.sa1
Author response https://doi.org/10.7554/eLife.79519.sa2

## Additional files

### Supplementary files
- MDAR checklist
- Supplementary file 1. Strain list.
- Supplementary file 2. List of classification models.

### Data availability
Software and documentation is fully available via Github. Data used for training classifiers is available on Zenodo. A demo detecdiv project that contains all information to train users on detecdiv is available on zenodo. All the links are provided in the manuscript file and dataset list.

The following datasets were generated:

| Author(s) | Year | Dataset title | Dataset URL | Database and Identifier |
| --- | --- | --- | --- | --- |
| Aspert T | 2021 | Annotated images from yeast cell lifespans - Training & Test sets - DetecDiv (id01) | https://doi.org/10.5281/zenodo.6078462 | Zenodo, 10.5281/zenodo.6078462 |
| Aspert T | 2021 | Trained network for classification of images from yeast cell lifespans - DetecDiv (id01) | https://doi.org/10.5281/zenodo.5553862 | Zenodo, 10.5281/zenodo.5553862 |
| Aspert T | 2021 | Annotated pixels from brightfield images from yeast cell in microfluidic traps - Training and Test sets - DetecDiv (id02) | https://doi.org/10.5281/zenodo.6077125 | Zenodo, 10.5281/zenodo.6077125 |
| Aspert T | 2021 | Trained network for segmentation of yeast cell from brightfield images in microfluidic traps - DetecDiv (id02) | https://doi.org/10.5281/zenodo.5553851 | Zenodo, 10.5281/zenodo.5553851 |
| Aspert T | 2021 | Annotated timeseries from yeast cell lifespans - Training and Test sets - DetecDiv (id03) | https://doi.org/10.5281/zenodo.6075691 | Zenodo, 10.5281/zenodo.6075691 |
| Aspert T | 2021 | Trained network for cell-cycle slowdown detection - DetecDiv (id03) | https://doi.org/10.5281/zenodo.5553829 | Zenodo, 10.5281/zenodo.5553829 |
| Charvin G, Aspert T | 2021 | DetecDiv Demo Project files | https://doi.org/10.5281/zenodo.5771536 | Zenodo, 10.5281/zenodo.5771536 |

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
