## [Editor Report]

In this work, the authors describe a novel method, based on deep learning, to analyze large numbers of yeast cells dividing in a controlled environment. The method builds on existing yeast cell trapping microfluidic devices that have been used for replicative lifespan assay. The authors demonstrate how an optimized microfluidic device can be coupled with deep learning methods to perform automatic cell division tracking and single cell trajectories quantification. The overall performance of the method is impressive: it allows to deal with large image datasets generated by timelapse microscopy several order of magnitudes faster than what manual annotation would require.

---

## [Decision Letter]

**Decision letter after peer review:**

Thank you for submitting your article "DetecDiv, a generalist deep-learning platform for automated cell division tracking and survival analysis" for consideration by *eLife*. Your article has been reviewed by 4 peer reviewers, one of whom is a member of our Board of Reviewing Editors, and the evaluation has been overseen by Carlos Isales as the Senior Editor. The following individual involved in the review of your submission has agreed to reveal their identity: Pascal Hersen (Reviewer #3).

In addition to the suggestions made by all the reviewers, which we hope are easily addressable, the essential revisions would include (for the authors):

1) Comparison with more recent Inception model, suggested by Reviewer 1.

2) Application of the proposed techniques to mutants and media conditions with known effects on survival rates and show that the results qualitatively match the expectations, as well as answering if similar conclusions are also be acquired with traditional/manual analysis (Reviewer 1).

3) Results of training the whole CNN+LSTM architecture in an end-to-end fashion, not separating training for two networks (Reviewer 2).

4) Comparison with the straightforward prediction of actual division times using that CNN+LSTM combination (Reviewer 2).

*Reviewer #1 (Recommendations for the authors):*

1. I understand that the study was performed from the point of an experimentalist who wants to obtain a functional tool. Still, I think the manuscript could be strengthened by including comparison with alternative choices for initial classification. For example, for the "Inception" CNN, two updates (Inception V2 and V3) have since been published by the authors. By testing different approaches, it may become apparent whether there is still (major) room for improvement.

2. The authors rigorously evaluate the performance of their approach compared to manual analysis, and even evaluate its performance on a broad range of data obtained with different experimental setups. To validate that the final, fully automated, pipeline can be used to extract meaningful biological information, they then apply it to mutants and media conditions with known effects on survival rates (Figure 2E and F) and show that the results qualitatively match the expectations. I think this could be further strengthened by comparing to what would be obtained with traditional 'manual' analysis. Would it be feasible to also create some ground-truth data for the mutants, to check whether the automated approach recapitulates not only qualitative but also quantitative effects? Similarly, a quantitative comparison to ground-truth analysis of Figure 5H and (at least parts of) Figure 6D would be good.

3. For the identification of the SEP, it is not entirely clear to me what the manual annotation for the ground-truth was based on. Is it only the duration of cell cycle phases? If so, would a simple 'algorithmic criteria' (such as a threshold cell cycle duration) recapitulating the manual choice then not achieve the same as the classifier?

4. To give a better assessment of the quality of the volume measurements based on semantic segmentation shown in Figure 4, it would be helpful to see the dependence of an object-wise accuracy as a function of the IoU threshold used to define correctly identified objects.

*Reviewer #2 (Recommendations for the authors):*

Even though the paper is well written, it does contain lots of information on all possible aspects of the study, which in my opinion slightly lacks the structure. Lots of ideas are presented, for example in terms of neural networks, but it reads like a report, where the authors just present chronologically how they developed the ideas. The same feeling is also about the experimental section, where experiments are just there to show what the pipeline can achieve and not to answer very well described initial research question(s). This is probably something which is difficult to address, but making some things more concise and more structured would help. Several ideas/descriptions/and statements are present in the main text (while describing those things) but then also in the discussion and methods. It is not exactly a repetition, which is good, but it is the same "information" that overlaps. For example the parts on the neural network are split between the Results and Method, and in both cases are very extensive.

Additionally to the above mentioned comments, a more interesting one is about the training of the CNN and LSTM. In the end the authors use both network and describe how they came to this decision, but why not show what the end-to-end training of the combined architecture CNN+LSTM would produce? It would be interesting how it compares to separate training, taking into account that having separate classification with a CNN and then do a kind of temporal analysis, independently from the first step, is suboptimal.

Another concern, which the authors can address is going back to prediction of actual division times. The authors argue in the beginning that "We selected this classification scheme – namely, the prediction of the budding state of the cell – over the direct assessment of cell division or budding (e.g., "division" versus "no division") because division and budding events can only be assessed by comparing successive frames, which is impossible using a classical CNN architecture dedicated to image classification, which takes a single frame as input. "

It was a good explanation in the beginning, before the authors went to LSTMs but then, in the end, they arrived at CNN+LSTM architecture that can easily predict the division times. Indeed the performance might be worse (or not) compared to DetecDiv, but even in that case, it is good to have a benchmark and show what the performance of such straightforward approach is, so to have a sort of a baseline.

---

## [Author Response]

In addition to the suggestions made by all the reviewers, which we hope are easily addressable, the essential revisions would include (for the authors):1) Comparison with more recent Inception model, suggested by Reviewer 1.

To address this request, for the division counting model, we have added two variants of the inception model, namely the inception v3 (Szegedy et al., 2015) and the latest inception-resnet v2 (Szegedy et al., 2016), in comparison to the original inception model v1 that we used in the manuscript (also known as GoogleNet, see Szegedy et al., 2014). Based on the training set and training procedures used in our study, we haven’t found any improvement neither with inception v3 nor with the larger inception-resnet v2 (see new Figure 2 —figure supplement 4 in the revised version). The agreement between ground truth and prediction is actually slightly less good than with the original inception model. This could be due to a tendency of the bigger models to overfit more.

However, because these two neural networks are bigger than the original inception one, we observed a ~3-fold (respectively ~6-fold) increase in classification times, see Figure

2 —figure supplement 4J. This increase in computing times is also reported here: https://fr.mathworks.com/help/deeplearning/ug/pretrained-convolutional-neural-networks. html.

Under these conditions, we think these more sophisticated models do not provide any substantial benefit for division counting yet are less practical to use for the developer and the end-user due to increased computing times. Nevertheless, these newer models are now fully compatible with the DetecDiv GUI and they could prove to be useful for future applications beyond the scope of this study.

2) Application of the proposed techniques to mutants and media conditions with known effects on survival rates and show that the results qualitatively match the expectations, as well as answering if similar conclusions are also be acquired with traditional/manual analysis (Reviewer 1).

We fully agree with the reviewer that this comparison is relevant to assess how well the model can deal with various genetic contexts that differ from the condition used for training. To answer this point, we have analyzed further the raw data to build ground truth (GT) datasets in the following contexts:

– Lifespan analysis: we have made a groundtruth (GT) with 35 cells for the *fob1delta* and *sir2delta* mutants and compared these manually annotated data to the network predictions – importantly, the network was only trained using WT data. The results indicate a very good agreement between prediction and GT, as displayed in the new Figure 2 —figure supplement 5A.

– We have manually annotated 35 cells grown in galactose to evaluate the agreement between GT and predictions using the same network as previously, which was trained on glucose data only. Here again, we observed a very good matching between the two, showing that the model can readily be applied to galactose data (and potentially other sugar sources) without further training, see new Figure 2 —figure supplement 5B-E.

– Stress survival analysis: we have made a GT with 35 cells to evaluate the time of death of cells exposed to 0.5mM H2O2 compared to the predictions of the classifier. The onset of cell death was evaluated by visual inspection by looking at the appearance (i.e., the refractive index, the presence of large vacuoles, etc.) of the cells. The comparison (i.e. the correlation between predicted death time and actual death time, and the confusion matrix for the time of death at the end of the experiment) is now displayed as a new Figure 6 —figure supplement 1. Here, again, there is a very good agreement between predictions and GT.

Overall, these new analyses suggest that the model is able to generalize very well when exposed to the new contexts that were investigated. Of note, the *sir2delta* mutant has a significantly bigger size than the WT strain, suggesting that the model is somewhat robust to cell size variations.

3) Results of training the whole CNN+LSTM architecture in an end-to-end fashion, not separating training for two networks (Reviewer 2).

To address the reviewer’s request, we have tried to train the CNN+LSTM in an end-to-end fashion. Yet, unfortunately, we have never been able to obtain an accuracy comparable to that obtained after separately training the CNN and the LSTM. Part of the reason was that training the network under these conditions was found to be very inefficient and demanding in terms of computing resources.

The LSTM training procedure uses feature vectors as inputs which are calculated only once for each sequence of images, based on the upstream CNN activations. In contrast, training the whole CNN+LSTM network uses image sequences as input that need to be processed by the whole architecture for each optimization step, hence is more intensive computationally. Also, because this method attempts to optimize CNN and LSTM layers weights within the same procedure, it is not clear how weights tuning is partitioned between the two parts of the network, knowing that they may have very different numbers of parameters and optimization requirements (the LSTM is naive, whereas the CNN is pre-trained).

Last, following a rapid survey of the literature, we point out that training video classification models based on such a CNN+LSTM architecture is usually performed using a sequential optimization of both parts of the network, as performed in our study.

4) Comparison with the straightforward prediction of actual division times using that CNN+LSTM combination (Reviewer 2).

We understand the point raised by the reviewer that an end-to-end training would alleviate the current limitation that we pointed out in the manuscript, namely, that the CNN cannot be trained to distinguish transitions between successive frames (i.e. onset of cell division or budding). However, as explained above, such an end-to-end training appeared to be quite unpractical, hence we did not proceed further to build a model that would attempt to distinguish ‘div’ vs ‘no div’ frames.

Reviewer #1 (Recommendations for the authors):1. I understand that the study was performed from the point of an experimentalist who wants to obtain a functional tool. Still, I think the manuscript could be strengthened by including comparison with alternative choices for initial classification. For example, for the "Inception" CNN, two updates (Inception V2 and V3) have since been published by the authors. By testing different approaches, it may become apparent whether there is still (major) room for improvement.

Please see above our detailed reply to the editor regarding this point.

2. The authors rigorously evaluate the performance of their approach compared to manual analysis, and even evaluate its performance on a broad range of data obtained with different experimental setups. To validate that the final, fully automated, pipeline can be used to extract meaningful biological information, they then apply it to mutants and media conditions with known effects on survival rates (Figure 2E and F) and show that the results qualitatively match the expectations. I think this could be further strengthened by comparing to what would be obtained with traditional 'manual' analysis. Would it be feasible to also create some ground-truth data for the mutants, to check whether the automated approach recapitulates not only qualitative but also quantitative effects? Similarly, a quantitative comparison to ground-truth analysis of Figure 5H and (at least parts of) Figure 6D would be good.

We have addressed this point above, see the point-by-point reply to the editor.

3. For the identification of the SEP, it is not entirely clear to me what the manual annotation for the ground-truth was based on. Is it only the duration of cell cycle phases? If so, would a simple 'algorithmic criteria' (such as a threshold cell cycle duration) recapitulating the manual choice then not achieve the same as the classifier?

Thanks for this suggestion. However, the main issue here is that the post-SEP period of the lifespan is characterized by large variability in cell cycle duration (see Fehmann et al., Cell Rep, 2013). Therefore, unfortunately, using simple thresholding to assess the position of the SEP may not be reliable, since a punctual long cell-cycle could be followed by another one that would fall below the threshold.

In this context, we have previously introduced a piecewise linear fit based on Chi2 minimization to determine the onset of the SEP (see Fehrmann et al., Cell Rep, 2013, see also Morlot et al., Cell Rep 2019), as shown in Author response image 1 (taken from Morlot et al., 2019).

**Author response image 1. sa2fig1:** 

This method, which looks somewhat reliable by visual inspection, may still strongly mislocalize the position of the SEP on some occasions (e.g. short post-SEP period, erratic post-SEP divisions, etc.). Therefore, we thought that using an appropriate classifier may be useful to overcome these limitations. In this context, We have generated the groundtruth by inspecting the cell cycle frequency vs time plot for each cell, using the piecewise linear fit as a visual aid.Beyond the determination of the SEP, which is quite successful, we believe that more generally, this classifier provides an interesting application of an LSTM classifier to identify a specific transition in image sequence datasets. In the revised version, we have added a sentence to better justify the need to use such an approach.

4. To give a better assessment of the quality of the volume measurements based on semantic segmentation shown in Figure 4, it would be helpful to see the dependence of an object-wise accuracy as a function of the IoU threshold used to define correctly identified objects.

Unless we misunderstand this statement, we think that such analysis is already present in Figure 5 —figure supplement 2A for cell contours and Figure 5 —figure supplement 3A for nucleus contours. These two figures display how precision and recall vary as we increase the ‘cell’ (and ‘nucleus’, respectively) class assignment threshold on the [0.2; 0.95] interval. Based on this analysis, we have chosen the threshold that maximizes the F1-score (Panel B in the same figures). We agree with the reviewer that this benchmarking provides a useful evaluation of the robustness of the classifiers.

Reviewer #2 (Recommendations for the authors):Even though the paper is well written, it does contain lots of information on all possible aspects of the study, which in my opinion slightly lacks the structure. Lots of ideas are presented, for example in terms of neural networks, but it reads like a report, where the authors just present chronologically how they developed the ideas. The same feeling is also about the experimental section, where experiments are just there to show what the pipeline can achieve and not to answer very well described initial research question(s). This is probably something which is difficult to address, but making some things more concise and more structured would help. Several ideas/descriptions/and statements are present in the main text (while describing those things) but then also in the discussion and methods. It is not exactly a repetition, which is good, but it is the same "information" that overlaps. For example the parts on the neural network are split between the Results and Method, and in both cases are very extensive.

In order to improve readability, we have made the following changes to the manuscript:

– We have moved some experimental/methodological details to the methods section whenever possible, in order to guide the reader through the main results of the paper. In this way, we have removed part of the redundancy pointed out by the reviewer.

– We have introduced one more section header to separate the hardware part from the description of the image processing pipeline.

– We have better justified the choices we made regarding the network architecture, and the procedures used throughout the paper.

We hope that these changes makes the revised manuscript easier to read.

Additionally to the above mentioned comments, a more interesting one is about the training of the CNN and LSTM. In the end the authors use both network and describe how they came to this decision, but why not show what the end-to-end training of the combined architecture CNN+LSTM would produce? It would be interesting how it compares to separate training, taking into account that having separate classification with a CNN and then do a kind of temporal analysis, independently from the first step, is suboptimal.

We thank the reviewer for this suggestion. We have addressed this point above in the reply to the editor (point #3).

Another concern, which the authors can address is going back to prediction of actual division times. The authors argue in the beginning that "We selected this classification scheme – namely, the prediction of the budding state of the cell – over the direct assessment of cell division or budding (e.g., "division" versus "no division") because division and budding events can only be assessed by comparing successive frames, which is impossible using a classical CNN architecture dedicated to image classification, which takes a single frame as input. "It was a good explanation in the beginning, before the authors went to LSTMs but then, in the end, they arrived at CNN+LSTM architecture that can easily predict the division times. Indeed the performance might be worse (or not) compared to DetecDiv, but even in that case, it is good to have a benchmark and show what the performance of such straightforward approach is, so to have a sort of a baseline.

We thank the reviewer for this suggestion. We provide a reply to this above in the reply to the editor (point #4).